# Cytomegalovirus and Inflammatory Bowel Diseases (IBD) with a Special Focus on the Link with Ulcerative Colitis (UC)

**DOI:** 10.3390/microorganisms8071078

**Published:** 2020-07-20

**Authors:** Alexandre Jentzer, Pauline Veyrard, Xavier Roblin, Pierre Saint-Sardos, Nicolas Rochereau, Stéphane Paul, Thomas Bourlet, Bruno Pozzetto, Sylvie Pillet

**Affiliations:** 1GIMAP EA 3064, Medicine Faculty of Saint-Etienne, University of Lyon, 69007 Lyon, France; alexandre.jentzer@gmail.com (A.J.); pauline.veyrard@chu-st-etienne.fr (P.V.); xavier.roblin@chu-st-etienne.fr (X.R.); nicolas.rochereau@univ-st-etienne.fr (N.R.); stephane.paul@chu-st-etienne.fr (S.P.); thomas.bourlet@chu-st-etienne.fr (T.B.); bruno.pozzetto@univ-st-etienne.fr (B.P.); 2Laboratory of infectious agents and hygiene, University Hospital Saint-Etienne, 42055 Saint-Etienne, France; 3Laboratory of Immunology, University Hospital Saint-Etienne, 42055 Saint-Etienne, 42055 Saint-Etienne, France; 4Department of Gastroenterology, University Hospital Saint-Etienne, 42055 Saint-Etienne, France; 5Laboratory of Bacteriology, University Hospital of Clermont-Ferrand, 63100 Clermont-Ferrand, France; saintsardos.pierre@gmail.com

**Keywords:** ulcerative colitis, CMV-associated colitis, T_H_2 cytokines, inflammation, tumor necrosis factor α, ganciclovir

## Abstract

Cytomegalovirus (CMV) infects approximately 40% of adults in France and persists lifelong as a latent agent in different organs, including gut. A close relationship is observed between inflammation that favors viral expression and viral replication that exacerbates inflammation. In this context, CMV colitis may impact the prognosis of patients suffering from inflammatory bowel diseases (IBDs), and notably those with ulcerative colitis (UC). In UC, the mucosal inflammation and T helper cell (T_H_) 2 cytokines, together with immunomodulatory drugs used for controlling flare-ups, favor viral reactivation within the gut, which, in turn, increases mucosal inflammation, impairs corticoid and immunosuppressor efficacy (the probability of steroid resistance is multiplied by more than 20 in the case of CMV colitis), and enhances the risk for colectomy. This review emphasizes the virological tools that are recommended for exploring CMV colitis during inflammatory bowel diseases (IBD) and underlines the interest of using ganciclovir for treating flare-ups associated to CMV colitis in UC patients.

## 1. Introduction

Inflammatory bowel diseases (IBDs) primarily involve two main entities: Crohn disease (CD) and ulcerative colitis (UC). IBDs affect mainly young and active subjects with a diagnosis generally made between 20 and 30 years old. Their annual incidence is increasing overall, with a relative stabilization in Western countries [1], but an accelerating incidence in newly industrialized countries, where the way of life is becoming “Westernized” [2]. The incidence is also increasing in adolescents, as reported by a recent French study [3]. These chronic diseases are incurable and disabling with a major impact on the socio-professional life of patients. Their pathophysiology is complex and dominated by chronic inflammation of the digestive tract, and involves genetic factors affecting innate and adaptive immune responses of the digestive mucosa together with environmental factors (Figure 1) [4,5,6]. IBDs are clinically characterized by the alternation of acute inflammatory flares and asymptomatic phases during remission [4,7]. The administration of anti-inflammatory or immunosuppressive drugs (steroids, azathioprine, cyclosporine, monoclonal antibodies especially directed against tumour necrosis factor α (TNF-α), or α4β7 integrin in the case of UC most often allows controlling the acute phase and lengthening the periods of remission. Several lines of treatment are sometimes necessary to achieve intestinal mucosa healing [8,9]. Despite these different therapeutic options, inflammation can lead to tissue necrosis requiring a partial or total colectomy.

Human cytomegalovirus (CMV), also termed human betaherpesvirus 5, is an opportunistic pathogen involved in many inflammatory processes [10]. The diagnosis of CMV colitis in IBD patients is difficult due to the absence of clinical or endoscopic symptoms that differentiate CMV colitis from colitis associated with the inflammatory disease itself; it is based exclusively on the detection of histological or viral markers in the intestinal mucosa [11,12]. The aim of this review was to highlight the place of colonic CMV infection during IBD by emphasizing the relationships between CMV infection of the colonic mucosa and inflammatory flare-up of UC, the virus and the inflammation interacting with each other, which results in perpetuating and worsening colonic lesions. We will also show how, in UC, immunosuppressive and antiviral treatments can help control the colonic inflammation.

## 2. Brief Recalls on the Pathophysiology of CMV Infection

Beyond these reminders limited to the objectives of this review, the reader interested in more information is invited to refer to recent general reviews on the subject [13,14].

### 2.1. Cell Tropism and Transmission of CMV

CMV belongs to the *beta-Herpesvirinae* subfamily within the *Herpesviridae* family. Ubiquitous around the world, it infects exclusively humans. The viral genome is a double-stranded DNA molecule protected by a capsid of icosahedral symmetry, a tegument, and an envelope. Fragile in the outside environment, CMV is transmitted through close contacts with secretions (saliva, milk, genital secretion, and semen) and biological fluids (urine) from an infected individual. At the time of primary infection, a viremia allows the virus to spread to all the organs; blood and organs are potential sources of iatrogenic transmission. During pregnancy, it can be transmitted from mother to foetus, CMV being the most common source of viral congenital infections (0.5–2%) of all live births and the main non-genetic cause of congenital sensorineural hearing loss and neurological damage [15]. CMV replicates in many cell types, including endothelial cells, epithelial cells, fibroblasts, and monocytes/macrophages.

### 2.2. Lytic Replication Cycle

In vitro, the viral cycle has mainly been studied in fibroblast cells. The attachment of a viral glycoprotein complex to cellular receptors (glycosaminoglycans like heparan sulphate, integrins, or even many growth factor receptors) allows the nucleocapsid to enter the cytosol and then enter into the cell nucleus. Viral genes’ expression progresses in three phases: The immediate early (IE) genes encode transcription factors that induce the expression of early (E) genes; these genes code for proteins involved notably in the replication of the viral genome, including viral DNA polymerase (pUL54) and thymidine kinase (pUL97). After replication of the viral DNA and expression of the structural or late (L) genes (capsid, envelope glycoproteins, and the tegument proteins), the viral genome is encapsidated. The nucleocapsid is matured during a complex pathway through cellular membranes to release new virions by budding. Transmission to a new cell takes place either through free virus particles or through intercellular contact.

### 2.3. Latency and Reactivation

Both innate and adaptive immunity are mobilized to control rapid CMV replication [13,16], with a crucial role for innate lymphoid cells (ILCs) and natural killer (NK) cells, production of neutralizing antibodies, and expansion of cytotoxic T lymphocytes. CMV then enters the latency phase, stopping the production of infectious particles and reducing viral expression to the proteins that maintain the latency program. This program is established for life in the body although the molecular and viral processes are still relatively unknown. Latent CMV infection is mainly observed in circulating hematopoietic CD34+ progenitors and monocytes, but other cells (notably endothelial cells), disseminated in the tissues, are probably susceptible to contain the latent genome in vivo [17]. During latency, viral DNA persists as an episome in the nucleus without integration into the cellular genome; viral expression is limited to specific proteins whose main role is to inhibit the presentation of viral epitopes to immune cells. Latent infection should be differentiated from persistence with low production of infectious particles, below the detection limit of standard techniques.

CMV can reactivate with the production of new infectious viral particles. The factors implicated are still poorly identified; stimulation of the immune system by infection, significant stress, inflammation, allogenic stimulations (pregnancy, transfusion, organ or hematopoietic stem cell transplant), or immunodepression (administration of immunosuppressive treatments and chemotherapy, human immunodeficiency virus (HIV) infection) can induce CMV reactivation. Reinfections with different strains are also possible.

### 2.4. Infection vs. Disease

In most cases, CMV infection is restricted by the immune system. However, CMV can affect the function of organs (brain, lung, digestive tract, etc.) leading to end-organ disease: In these cases, the term of CMV disease is to be used and an antiviral therapy should be administrated to hamper life-threating disease [18]. CMV disease mainly occurs in immunosuppressed patients but cases are reported even in immunocompetent patients, especially after primary infection. In the case of CMV disease, CMV replication markers are detected in the infected organ; viremia can be absent, especially when the reactivation occurs primarily in the organ before disseminating to the peripheral blood. CMV colitis in UC patients should be considered as a CMV disease.

### 2.5. CMV and Inflammation

The interactions between CMV and the immune system are complex, both the actors of innate immunity and those of the adaptive response [10,16,19,20]. CMV is often considered as an “immunopathogenic” virus [10]. Activation of the immune system begins very early, activated after infection, upon recognition of viral proteins by toll-like receptors (TLR), or activation of type I interferons (IFNs) [20]. CMV infection increases the secretion of numerous cytokines including transforming growth factor β (TGF-β), TNF-α, interleukin 1β (IL-1β), IL-6, platelet-derived growth factor (PDGF), regulated on activation, normal T cell expressed and secreted (RANTES), monocyte chemoattractant protein 1 (MCP-1), macrophage inflammatory protein 1α (MIP-1α), and MIP-1β [20,21]. Recognition of infected cells by cytotoxic cells also induces the production of pro-inflammatory cytokines [20]. A population of unconventional T lymphocytes, Tγδ lymphocytes, plays a preponderant role during CMV infection and is notably present in the intestinal tissue. In the case of CMV infection, they are able to induce an early synthesis of pro-inflammatory cytokines and to develop their cytotoxic activity [22]. A major evasion mechanism is the inhibition of human leucocyte antigen (HLA) class I-restricted antigen presentation [16]. The processing and the presentation of CMV antigenic peptides via the HLA class I pathway was blocked to prevent specific lymphotoxicity. Several viral proteins block HLA class I-peptide complexes’ trafficking and induce a rapid down-regulation in HLA class I expression. Antigen presentation through the HLA class II pathway is also hindered by CMV. This down-regulation of HLA molecules exposes them to NK-mediated lysis; in counterpart, CMV develops various tactics to impede NK cell recognition. CMV also encodes a variety of other cytokines’ homologues (IL-10) with distinct subversive functions and that mimic the behaviour of host proteins to divert the immune response [16]. The virus also activates the expression of inflammatory cytokines through the stimulation of the transcription factor nuclear factor-kappa B (NF-κB), cyclooxygenase-2, and 5-lipooxygenase [23]. All these pro-inflammatory mediators participate in the activation of viral replication. The development of a vicious circle between CMV infection and inflammation has been reported [10,21,24]. Because of the close interactions between CMV and the immune system, host genetics are probably implicated in the virus pathogenesis [25], although no routine test is currently available for predicting the risk of severe or recurrent infection.

### 2.6. Antiviral Drugs against CMV

The main antiviral molecule used in prophylactic or curative treatment, especially in immunocompromised patients, is ganciclovir (GCV) and its orally administered prodrug, valganciclovir. This analogue of guanosine is an inhibitor of the viral polymerase (pUL54), acting as a chain terminator [26]. In the case of resistance to GCV, two others inhibitors of pUL54 can be used, although more toxic for kidney functions: Foscarnet, a structural analogue of the pyrophosphate anion that selectively inhibits the pyrophosphate-binding site on viral DNA polymerase, or cidofovir that competes with viral polymerase as an analogue of cytidine. Letermovir, a new antiviral drug that inhibits pUL56 subunit during CMV particles’ maturation [27], is now used as preventive therapy in hematopoietic stem cells (HSCs) receivers [28]. Other molecules (cidofovir esters, maribavir, artesunate, etc.) are arriving on the market; however, their indications remain presently extremely limited. There is currently no CMV vaccine, although many teams are working either to prevent primary infection or to limit the consequences of the infection, particularly in immunocompromised patients [29].

## 3. CMV Infection of the Digestive Tract in Immunocompromised and Immunocompetent Patients

CMV seroprevalence is estimated to be around 66% of the adult population in Western countries; a higher seroprevalence is observed in low-economic countries [30]. CMV seropositive people harbor latent CMV genome in many tissues, including the entire digestive tract; the colon is considered a major site for latency and viral reactivation. Viral inclusions are often found in endothelial cells but also in macrophages, in epithelial cells, in fibroblasts, and in smooth muscle cells [31,32]. The mechanism of the pathogenesis of CMV colitis remains poorly understood; replication of the virus in endothelial cells could generate vasculitis associated with microvessel thrombosis and local ulceration, which would induce ischemic colitis [33,34]. The recruitment to the mucosa of CMV-infected monocytes promotes the dissemination of inflammatory macrophages in the inflamed tissue [35,36]. CMV also disrupts epithelial tight junctions, enhancing bacterial translocation as well as inflammation of the gut [37].

In immunocompetent patients, several cases of digestive damages have been reported in primary infection, ranging from esophagitis to proctitis (excluding IBD) [32,38]. The diagnosis is often evocated late, after performing an endoscopy with detection of viral markers in the biopsy, leading to delay the antiviral treatment, which may engage vital prognosis [39]. In addition, digestive damages are observed in newborns with congenital CMV infection and can be complicated by necrotizing enterocolitis [40].

In immunocompromised patients, the digestive tropism of CMV is expressed by the frequency of digestive infections, particularly in recipients of stem cells or organ transplant and in HIV-infected patients infected at AIDS stage [16,19,24,32,41,42]. In addition, CMV infection is a major cause of graft rejection after intestinal transplantation [10].

## 4. Controversial Role of CMV Infection in IBD Inflammatory Flares—Analysis of Confounding Factors

In the absence of a specific study of CMV seroprevalence in IBD patients, it can be reasonably considered to be at least equivalent to that of the general population. In moderate to severe UC adult patients, CMV colitis is observed in 25–30% [43,44,45], especially those exhibiting steroid refractory disease [46]. The detection of CMV in colonic mucosa is less frequent in IBD children than in adults [47], probably because of lower seroprevalence of CMV infection in the pediatric population [48] and lower risk of viral reactivation; however, the consequences of CMV colitis on IBD evolution is similar to that observed in adults [49,50]. CMV colitis can reveal the inflammatory disease which was not diagnosed until the viral episode [51]. In contrast, CMV infection is not considered to be an etiology of the inflammatory disease (Figure 1). Reactivated by damage to the intestinal mucosa and local inflammation, it can promote flare-ups, exacerbate mucosal damages, and shorten the periods of remission.

During the last 10 years, the role of CMV in IBD flare-ups has been highly debated, some authors favoring an active role of the virus in inflammatory flares whereas others promoted the concept of the innocent bystander in which CMV infection plays no role in the progression of the disease [46,52]. A meta-analysis published in 2006 showed a prevalence of CMV infection in IBD patients ranging from 0.5 to 100% [53]. Two studies on UC patients published 10 years ago illustrate this controversy: The first reported an identical colectomy rate between CMV-positive and CMV-negative patients, and spontaneous disappearance of markers of infection in CMV-positive patients [54], while the second showed an association between CMV infection and steroid resistance [55]. These discrepancies can be explained by the inclusion of patients with different inflammatory diseases (UC versus CD), with heterogeneous clinical scores and treatments, or using inappropriate CMV detection methods.

### 4.1. Differences between Primary Infection and Tissue Reactivation

It is important to distinguish the specific case of primary CMV infection in patients with an established IBD (UC or CD) and treated with immunosuppressive therapy (especially azathioprine). In this context, the clinical presentations of CMV primary infection associate a significant viremia with tissue damage, pneumonia, and hemophagocytic lymphohistiocytosis. CMV primary infection can lead to the patient’s death, even if ganciclovir is quickly introduced [46]. This risk of primary infection is high in low-seroprevalence IBD populations [56]. Digestive damages are rarely described in this case and were excluded from the scope of this review.

### 4.2. UC versus CD

Anatomically, UC and CD affect the digestive tract in different ways: In UC, the damage of the rectum is constant, together with variable and continuous lesions of the colon, whereas in CD the lesions are discontinuous throughout the whole digestive tract. From an immunological point of view, CD is characterized by a T helper cell T_H_1/T_H_17 with production of IFN-γ exhibiting antiviral properties and a cytotoxic T cell (T_c_) activation [20], whereas, in UC, the T_H_2/ T_H_9 profile does not prevent CMV replication [57] (Figure 2). These major immunological differences could explain why CMV tissue reactivation is rare in CD flares whereas it is frequent in UC patients. In our cohort, we showed that CMV tissue infection was of 11% in steroid refractory CD patients versus 38% in UC patients [58]; similar differences have been reported in recent reviews [46,59].

### 4.3. Virological Diagnosis of CMV Colitis

CMV mucosal infection cannot be diagnosed by endoscopic examination of the mucosa alone and viral markers should be analyzed in the clinical specimens [12,62]. The reported methods for detecting CMV infection are very heterogeneous and constitute a major bias in results’ interpretation. CMV infection can be demonstrated by using either indirect methods (detection of specific immunoglobulins IgM and IgG) or direct ones (detection of virus or its components). Some of these methods (reviewed in [36,46]) are not suitable for assessing the tissue reactivation of CMV from its intestinal reservoir.

The presence of IgG against CMV allows targeting patients already exposed to CMV, i.e., at risk of viral reactivation. IgM and IgG serology can also be used to demonstrate a primary infection of unfavorable prognosis. Apart from these indications, serology is not of interest for monitoring patients with IBD.

Except in primary infection, viremia is uncommon in UC patients [44,63,64] and only occurs after several days or weeks of CMV replication in inflammatory colonic tissue. Consequently, viremia can be considered as a marker of less favorable prognosis. In contrast, in order to monitor immunosuppressed patients, techniques that aim to detect CMV in peripheral blood (pp65 antigenemia and viral load by qPCR) are not sensitive enough to early assess CMV colitis [41,65,66]. Studies supporting the role of CMV as innocent bystander have globally detected the stigma of CMV infection in patient’s blood by using pp65 antigenemia or serological techniques and have rarely focused on the presence of CMV in gut [46].

As CMV reactivation initially localizes in the colon of UC patients, the techniques detecting CMV markers in the tissue are the most relevant [44,66] and are currently recommended by guidelines [67,68]. Histological examination with observation of “owl-eye” cells after hematoxylin-eosin (H&E) staining reflects the CMV in vivo cytopathogenic effect that is pathognomonic of the tissue infection. In a recent study comparing different methods, the presence of “owl-eye” cells is of bad prognosis with increased risk of colectomy in IBD patients [69,70]. However, histopathological analysis alone is not recommended to assess CMV colitis in IBD patients [66,71]. In order to increase the sensitivity of this technique [72], cells expressing immediate early (IE) and early (E) virus antigens can be revealed by immunohistochemistry (IHC); this technique also allows a semi-quantification of viral infection after counting the number of colored nuclei by field under an optical microscope [73,74].

Molecular methods that measure tissue viral load by qPCR are more and more used. Nested PCR assays should be prohibited due to the high risk of contamination generating false-positive results. Fresh biopsies should be preferred to formalin-fixed and paraffinized ones, the treatment for histological examination reducing the integrity of nucleic acids and consequently the sensitivity of the molecular assay. International guidelines recommend IHC or tissue qPCR [67,68], both being the techniques currently used by various teams. The qPCR is more sensitive than IHC [73,74,75,76], more reliable, and independent of the observer [77]. This molecular technique may however give “false-positive” results in the case of low viral load. It does not assess the infectious character of the detected genome, whereas observation of infected cells expressing viral antigens does.

The definition of viral load thresholds for therapeutic management of patients is urgently needed [26,41,44,48]; a consensus on results’ expression using qPCR must also be established since each team uses different units (Table 1), which makes it difficult to compare results [78,79,80,81].

The mucosal site to sample is also questionable. CMV infection markers are not detected in healthy tissue; they are found only in inflammatory tissue [44,74,80]. Different authors recommend analyzing several biopsies because (1) the inflammation is distributed heterogeneously in tissue, (2) the analysis is carried out on a tiny sample of mucosa and, (3) it is not easy to assess the inflammatory nature of the mucosa during endoscopic observation [45,74,81,89,90]. The observation of ulcers that represent particularly inflammatory areas of the digestive mucosa is correlated with CMV tissue infection [74,91]. These sites, when present, must be preferred to detect markers of CMV colitis [11,46,74,92].

### 4.4. Typing the CMV Strains

As the patient can be infected by several genotypes, some of them could be associated to the severity of the disease. Only a few studies have depicted such relation [93,94]; more data are needed, especially around the world as some genotypes are found mainly in Asian countries.

In the same interest, the sensitivity of CMV to antiviral drugs has been poorly studied in biopsy specimens, and only in transplant patients [95], the techniques of sequencing being less sensitive than qPCR. 

### 4.5. Other Assays for CMV Colitis Diagnosis

In the future, non-invasive approaches should be evaluated. Among them, the measurement of CMV peripheral T-cell response needs assessment in UC patients [63,96,97]. To date, QuantiFERON-CMV assay or Enzyme-linked immunospot (ELISPOT) assay that measures IFN-γ level produced in vitro by circulating T cells in response to the stimulation with CMV antigens has been used to evaluate the risk of CMV reactivation in stem cells and organ transplant patients [98,99]. These assays could help to classify UC patients at low or high risk of viral reactivation.

Detection of CMV DNA in stool specimens could also be an alternative to biopsy examination [94,100,101,102,103]; but, to date, the sensitivity of the assays is too low, leading to false-negative results, even in allograft patients with high viral load in the tissue [104].

## 5. Inflammation and Immunosuppressive Therapies Contribute to Colonic Reactivation of CMV Infection during UC

In UC, the mucosa is fragile, with erosions, bleeding, and even superficial or deep ulcers of inflammatory areas. Disease activity is assessed by a Mayo clinical score [105], associated with a mucosa inflammatory state estimation during endoscopy by an endoscopic Mayo subscore [106]. These scores are also used to monitor the efficacy of drug treatment. CMV reactivation in the intestinal mucosa, highly related to the degree of mucosal inflammation, is observed primarily in inflammatory or ulcerated areas. UC patients have an alteration of their digestive mucosa with significant inflammation and even ulcerations. In addition, they receive immunosuppressive treatments. Therefore, they combine two major risk factors promoting CMV reactivation.

### 5.1. Impact of Inflammation

UC affects the digestive mucosa, with an alteration of the intestinal barrier and microbiota, an exacerbated reaction against digestive tract antigens, including nonpathogenic commensal bacteria, and an alteration of innate and adaptive immune responses, resulting in the local production of many inflammatory cytokines such as TNF-α, IL-6, and IL-23 (Figure 2B). TNF-α is capable of recruiting reservoir cells harboring latent virus such as monocytes that differentiate into macrophages and produce viral infectious particles [36,107]. The binding of TNF-α to its receptor activates protein kinase C and NF-κB pathway which, in turn, stimulates the transcription of the IE genes and, thus, viral replication [10,16]. Other mediators such as pro-inflammatory prostaglandins, stress catecholamines, epinephrine, and norepinephrine also activate the expression of IE genes by increasing cyclic adenosine monophosphate (cAMP) [16]. In UC patients, CMV replication is, thus, particularly favored by chronic inflammation [21].

### 5.2. Anti-Inflammatory and Immunosuppressive Drugs

Table 2 summarizes the recommendations of UC treatment according to recent national guidelines [68].

The reactivation of CMV is promoted by the administration of drugs with immunosuppressive properties by systemic route [108,109]. Corticosteroids are the first line of treatment for moderate to severe IBD flare-ups but they promote CMV reactivation [23,43,46,55,108,109]. In a recent meta-analysis of 16 observational studies, corticosteroid exposure doubled the risk of CMV tissue reactivation (odd ratio (OR) = 2.10, 95% confidence interval (CI) = 1.31–3.37) [109]. Corticosteroids generally promote infections by down-regulating monocytes’ and T lymphocytes’ activities. In addition, corticosteroids facilitate IE genes’ transcription [110] and reactivation of latent virus in infected cells [111]. In turn, it has also been proposed that CMV alters the expression of glucocorticoid receptors (GR), with an increased ratio of GR β/α and phosphorylation of the α isoform, leading to refractory response to these immunomodulator drugs during lytic infection [112].

The risk of CMV infection in patients with UC treated with thiopurines is also increased (OR = 1.76, 95% CI = 1.21–2.57) [109]. Cyclosporin A administration was also shown to promote CMV reactivation, although these two series included a limited number of included patients [55,113].

Conversely, monoclonal antibodies against TNF-α such as infliximab or adalimumab do not increase the risk of CMV reactivation [44,62,66,67] and tissue CMV infection is not associated with clinical resistance to these biotherapies [114]. These beneficial results of anti-TNF-α on CMV infection are probably related to a reduction of TNF-α pro-inflammatory effect and a decrease in viral replication promoted by this cytokine [107]. Consequently, we have proposed to favor these biotherapies in the treatment of moderate to severe flare-up associated to CMV colonic infection [46].

Concerning vedolizumab, a new immunosuppressive drug that targets the homing of α4β7 lymphocytes, a risk of severe CMV infection has been suggested in the context of either primary infection [115] or reactivation [116,117]. Other studies are needed to confirm these preliminary observations. When this drug is used, it may be prudent to add antiviral effective against CMV, at least in patients at risk of CMV colitis.

Tofacitinib is a recent oral Janus kinase inhibitor approved for the treatment of patients with UC in several countries. Little information regarding CMV reactivation induced by this treatment is available. A review focused on the safety of tofacitinib for treatment of UC reported only one case of CMV colitis [118]. The in vitro incubation of peripheral blood mononuclear cells from healthy CMV-seropositive donors with tofacitinib and CMV antigens was less inhibitory against CMV-specific cytokines compared to tacrolimus [119]. The number of CMV-specific IFN-γ-producing cells was also modestly decreased by day 15 in CMV-seropositive healthy volunteers who received oral tofacitinib [120]. These data suggest that tofacitinib is a safe treatment regarding the risk of CMV infection.

Ustekinumab, a human monoclonal antibody directed to both anti-IL-12 and anti-Il-23, does not have, to date, approval for UC patients [68]. However, in a cohort of highly refractory UC patients with multiple prior drug failures, ustekinumab provided steroid-free clinical remission in one-third of cases at weeks 12–16 [121]. CMV infection was not recorded in this study.

## 6. Active Pejorative Role of CMV Reactivations in UC Flare-Ups

Once the various confounding factors have been eliminated, a consensus is emerging in the gastroenterology community that CMV colonic reactivation constitutes a factor of poor prognosis for moderate to severe flare-up of UC [46]. Table 3 summarizes the arguments that underline the pejorative impact of CMV colonic reactivation on UC evolution.

Consequently, international recommendations mention for several years the need for detection of CMV markers in the case of resistance to different lines of treatment and, in the case of positivity, the introduction of antiviral drug without interrupting the immunosuppressive therapy [67,68].

### 6.1. Association of CMV Infection with a Pejorative Evolution of UC

CMV infection in UC flare-ups is a major factor of nonresponse to immunosuppressive therapy, particularly to corticosteroids [72]. It is also associated with a higher rate of colectomy [55,79,86] and a risk of superinfection with *Clostridium difficile* [148,149,150]. In addition, these risks increase the length of patient hospitalization [1,91,150,151].

### 6.2. Association with Steroid Resistance and Resistance to Immunosuppressive Treatments

The use of steroids is considered as a major risk factor [46,109] (Table 3). CMV colitis is reported in about 30% of steroid-resistant UC flare-ups [43,44,45]. By using qPCR, we established a viral load threshold associated with steroid resistance of 10 copies/mg of tissue [44] or 5 international units (IU) per 100,000 cells [81] (Table 1).

We also showed that tissue CMV viral load predicts the response to several lines of immunosuppressive treatment: A viral load greater than 250 copies/mg of tissue or 370 IU/100,000 cells (sensitivity 95.2% (95% CI: 88.1–100%) and specificity 97.2% (95% CI: 93.5–100%)) is associated with resistance to more than two successive lines of treatment [44,81]. In contrast, as mentioned above, tissue CMV infection does not alter the efficacy of anti-TNF agents, even if the viral load is high [114].

### 6.3. Indications of Viral Load Measurement During UC Flare-Ups

Several authors propose measuring viral load by IHC or qPCR to adjust anti-inflammatory treatment and to consider antiviral therapy [11,46,152,153,154]. Currently, the heterogeneity of methods and thresholds that are used for identifying CMV reactivation does not make possible the establishment of consensual values for therapeutic adjustment (Table 1). In addition, the importance of multiple biopsies has been emphasized to avoid false-negative results, especially when the estimation of the mucosa inflammatory state is difficult during endoscopy [81]. Despite these technical difficulties, it is now clear that patients exhibiting CMV markers at the gut level should receive antiviral therapy.

As it seems difficult to perform rectosigmoidoscopy with inflammatory tissue sampling to all patients with a flare-up of UC, current studies are focusing on defining predictive factors able to target patients for whom CMV research in inflammatory tissue would have the best cost/benefit ratio (Table 4).

One of the first and easy analyses is to screen UC patients with CMV serology in order to propose CMV detection only in IgG-positive ones [12,156]. Several other criteria have been proposed but there is currently no consensus on any of them, except the presence of ulcers on endoscopy should encourage analysis of viral markers in inflammatory tissue.

### 6.4. Place of Antiviral Treatment

In the literature, several publications report isolated cases or small series of patients exhibiting UC relapses associated with CMV colitis, who have been treated with ganciclovir (or valganciclovir). Antiviral therapy is the most appropriate one for moderate to severe, steroid-refractory relapses with a high viral load [79]. A meta-analysis confirmed a decreased risk of colectomy (OR 0.20; 95% CI 0.08-0.49) when antiviral therapy is introduced in the case of steroid refractory flare-ups associated to CMV colitis [152]. We showed that ganciclovir treatment of few UC patients resistant to several therapeutic lines resulted in negativation of the DNA viral load in the inflamed tissue and restored the response to immunosuppressive therapies [44]. This success has been confirmed in other case reports or analysis of limited cohorts of patients (Table 3). The duration of antiviral treatment should be at least of two weeks and, if possible, by intravenous route [157], as the inflammation of the gut may compromise the drug absorption. For the management of such patients, we proposed an algorithm that considers the CMV viral load in inflammatory tissue (Figure 3). 

For most authors, antiviral treatment should not suspend immunosuppressive therapy in order to obtain a synergistic effect on inflammation and viral replication [116,159], especially when anti-TNF-α therapy is used [107,114]. The European Crohn’s and Colitis Organisation (ECCO) recommends continuing the immunosuppressive therapy in the case of CMV colitis and to stop it in the case of systemic infection [67]. The British Society of Gastroenterology recently published that “CMV reactivation in the colonic mucosa of patients hospitalized with an exacerbation of UC may be treated by intravenous ganciclovir 5 mg/kg twice daily while continuing conventional therapy with corticosteroids or rescue medication with infliximab or ciclosporin” [68]. A repeat endoscopic procedure may be useful to control the negativation of colon viral load.

### 6.5. Other Therapeutic Options

Some authors propose to use granulocyte/monocyte adsorptive apheresis [160,161,162,163,164], especially in Japan. Tacrolimus has also been proposed as an alternative solution to care for UC patients with CMV colitis [54,87,131].

## 7. Conclusions and Perspectives

UC therapeutic management is complex and depends on disease activity, location, progression, extraintestinal manifestations, and side effects of the immunosuppressive drugs. Algorithms have recently been published to better rationalize this management [7,12,46,156,165,166]. In addition to these recommendations, we suggest that anti-TNF biotherapy, combined with a ganciclovir-type antiviral agent, should be preferred in the case of a moderate or high viral load. This management strategy should be confirmed by a randomized trial including patients with UC flare-ups of steroid-refractory UC associated with CMV colitis.

Most studies have focused on CMV infection. Other viruses, particularly of the *Herpesviridae* family, may also play a role in UC flare-ups, notably in association with CMV. For example, co-infection of CMV with Epstein Barr virus (EBV) or human herpesvirus-6 (HHV-6) was shown to be an independent risk factor for colectomy [167]. In addition, high viral loads for EBV have been found in inflammatory tissue, especially in refractory disease [80]; it cannot be excluded that its presence may be associated to negative evolutionary factors, particularly that of digestive lymphoma. The enteric virome has also been shown to be altered in IBD [168], which could enhance the dysbiosis and the inflammation.

Back to CMV, a better understanding of the complex interactions between the local immune system and the colonic viral infections would make it possible to envisage better targeted therapeutic strategies, including therapeutic vaccination to “cure” the intestinal viral reservoir.

## Figures and Tables

**Figure 1 microorganisms-08-01078-f001:**
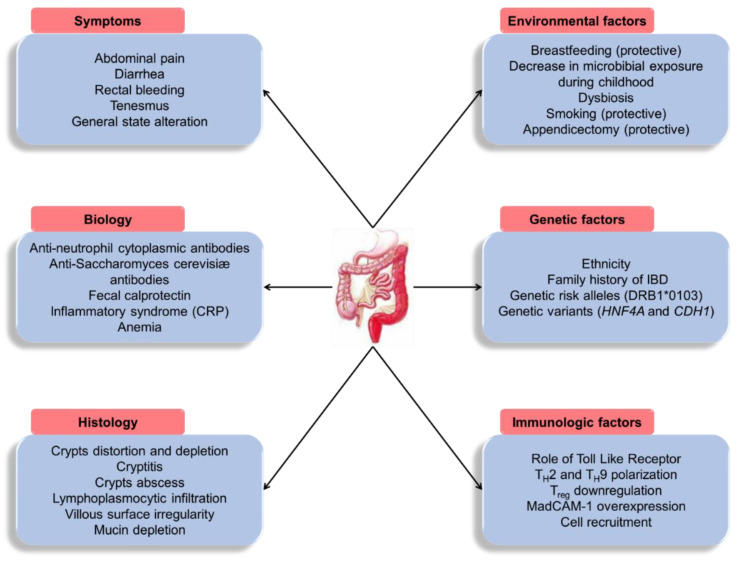
Main characteristics and risk factors of ulcerative colitis. CRP: C-reactive protein; IBD: Inflammatory bowel disease; T_H_, T helper cell.

**Figure 2 microorganisms-08-01078-f002:**
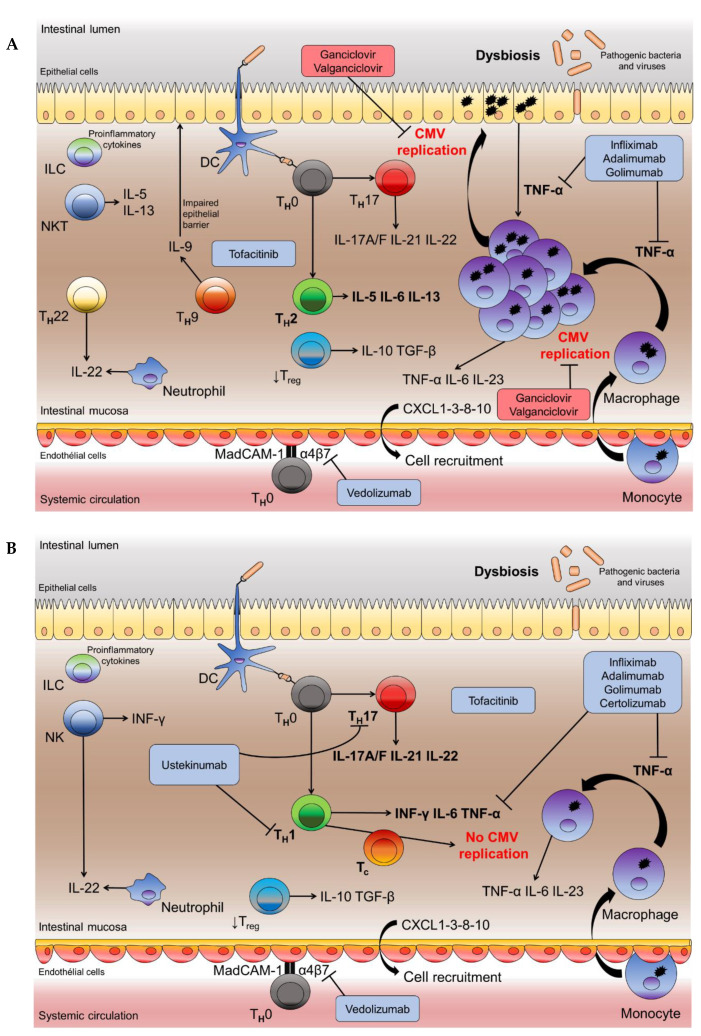
Pathophysiological hypothesis of ulcerative colitis and Crohn disease in the context of cytomegalovirus (CMV) viral reactivation. The rupture of the epithelial junctions increases intestinal permeability and facilitates the passage of microorganisms from the intestinal lumen to the lamina propria. Microorganisms are processed by dendritic cells (DCs) and macrophages through toll-like receptor (TLR). Antigens are primed by human leukocyte antigen (HLA) type II to be presented to T helper cells (T_H_) by TCR (T cell receptor). In the presence of pathogenic bacteria and viruses (dysbiosis), MAMPs stimulate the production of pro-inflammatory cytokines from epithelial cells, DCs, and macrophages. Activated macrophages synthesize pro-inflammatory cytokines such as TNF-α, IL-6, and IL-23. ILC, NK, and NKT also participate in pathophysiology of IBD. Tregs have a reduced functionality that does not allow them to counteract the local inflammatory state. Chemokines such as CXCL1-3-8-10 allow cell recruitment, which amplifies local inflammation. MadCAM-1 and α4β7 integrin are important molecules for the passage of lymphocytes into the lamina propria. The hematopoietic cells, especially monocytes under the effect of pro-inflammatory cytokines (TNF-α) and chemokines, are recruited from the intestinal mucosa, where they differentiate into macrophages. (**A**) In ulcerative colitis (UC), T_H_ differentiate into T_H_2 with secretion of IL-5, IL-6, and IL-13 and into T_H_17 with secretion of IL-17 A/F, IL-21, and IL-22. IL-9 produced by T_H_9 cells inhibits cellular proliferation and repair and has a negative effect on intestinal barrier function. IL-22 produced by T_H_22 allow the epithelial proliferation. T_H_2 and T_H_9 polarization of ulcerative colitis would allow viral reactivation and multiplication. Then infection of intestinal cells such as epithelial cells would allow TNF-α production and induces an inflammatory amplifying loop promoting viral replication. (**B**) In Crohn disease (CD), T_H_ differentiate into T_H_1 with secretion of INF-γ, IL-6, and TNF-α and into T_H_17 with secretion of IL-17 A/F, IL-21, and IL-22. T_H_1 and T_H_17 polarization of Crohn disease would allow better activation of Tc and would prevent CMV replication by controlling the reservoir. The treatments shown in the figure have received approval in France for IBD. The monoclonal antibodies anti-TNF-α are commonly used in IBD: Infliximab, Adalimumab, and Golimumab are used for both CD and UC. Certolizumab is only recommended for CD patients. These treatments inhibit a key player promoting inflammation, the TNF-α. Vedolizumab is a humanized monoclonal antibody to the α4β7-integrin that blocks lymphocyte trafficking to the gut and is approved for treatment of patients with CD and UC. Ustekinumab, a human anti-IL-12 and anti-Il-23 monoclonal antibody, has been indicated in CD. JAKs (Janus kinase) are members of intracellular non receptor tyrosine protein kinases that convert extracellular signaling into a wide range of cellular processes, including immune and inflammatory responses (JAK/STAT pathway). Tofacitinib blocks JAKs-mediated inflammatory pathways and modulates the adaptive and innate immune responses involved in IBD pathogenesis. Adapted from [5,60,61]. Abbreviations: CMV: Cytomegalovirus; CXCL: Chemokine (C-X-C motif) ligand; DC: Dendritic cell; IL: Interleukin; ILC: Innate lymphoid cell; INF-γ: Interferon gamma; JAK: Janus kinase; MadCAM-1: Mucosal addressin cell adhesion molecule 1; MAMPs: Microbe-associated molecular pattern; NK: Natural killer cell; NKT: Natural killer T cell; NOD2: Nucleotide-binding oligomerization domain 2; Tc: Cytotoxic T cell; TGF-β: Transforming growth factor beta; T_H_: T helper cell; TNF-α: Tumor necrosis factor alpha; Treg: Regulatory T cell.

**Figure 3 microorganisms-08-01078-f003:**
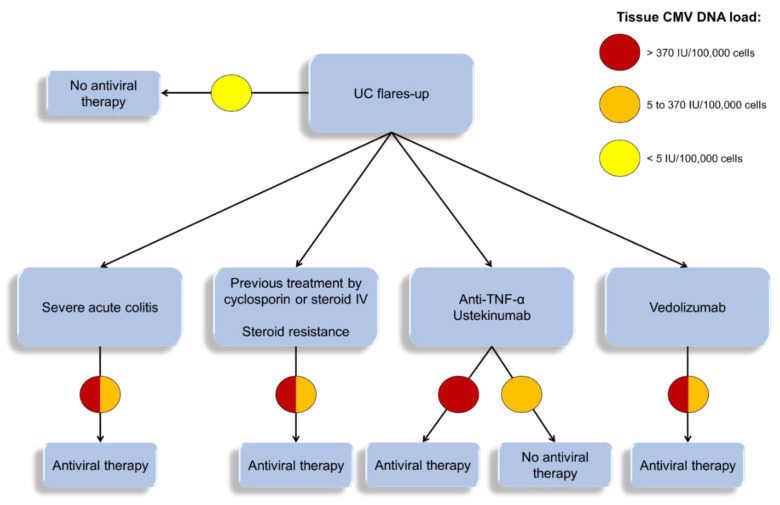
Therapeutic algorithm for the intake of flare-ups of ulcerative colitis (UC), according Table 1. The algorithm is drawn according to the last recommendations [8,68,158] and publications [44,46,81,114,116,159,160,161]. CMV: Cytomegalovirus; IV: intravenous; TNF: Tumor necrosis factor; UC: ulcerative colitis.

**Table 1 microorganisms-08-01078-t001:** Thresholds proposed in the literature to consider CMV infection to be of significant impact.

Method	Number of patients	Threshold	Reference
**IHC**			
	80 UC	0.01 cells/mm^2^	[82]
	114 UC	1 inclusion body in histology, positive specific IHC staining for CMV in 10–20 fields of colonic biopsies	[55]
	13 CD, 18 UC	semiquantitative ranging from 1 to 3 (1=rare, 2=common and easy to find, and 3=numerous cells positive for CMV)	[83]
	30 UC, 8 CD	semiquantitative ranging from 1 to 3 (1 = rare, 2 = common and easy to find, and 3 = numerous cells positive for CMV)	[84]
	95 UC	≥ 5 IHC positive cells/biopsy	[85]
	46 UC	colectomy was only associated with > 2 positive cells/biopsy	[86]
	257 UC	semiquantitative low grade (1 to 4 inclusion bodies per section) high grade (at least 5 inclusion bodies per section)	[70]
**qPCR**			
	30 UC	10 copies/µg of DNA	[87]
	42 UC	10 copies/mg of tissue	[44]
	40 IBD	> 1000 copies/100,000 cells	[80]
	149 UC	5 IU/100,000 cells	[81]
	92 UC, 9 CD	≥ 316 copies/mg of DNA	[64]
	75 IBD	10,000 copies/µg of DNA	[88]
	103 UC, 30 CD	> 600 copies/100 000 cells of tissue	[45]
	47 IBD	> 250 copies/mg of tissue	[42]

IHC: Immunohistochemistry, qPCR: Quantitative polymerase chain reaction, UC: Ulcerative colitis, CD: Crohn disease, IBD: Inflammatory bowel disease, IU: International unit.

**Table 2 microorganisms-08-01078-t002:** Medical treatments used in ulcerative colitis according to 2019 consensus British Society of Gastroenterology guidelines [68].

Treatment	Route of Administration	Mechanism of Action	Recommendations
**Salicylates**			
5-ASA	Oral	Decrease inflammation by blocking cyclooxygenase and inhibiting the production of prostaglandins. Inhibition of the NFкB pathway (decrease in the production of pro-inflammatory cytokines).	Initial treatment of mild to moderate UC should be treated with oral 5-ASA. Oral 5-ASA should be the standard maintenance medical therapy in ulcerative colitis.
Corticosteroid therapy			
	Oral IV	Decrease of chemotaxis towards inflammatory sites. Inhibition of microbicidal and phagocytosis functions. Inhibition of T lymphocyte functions	Moderate to severe ulcerative colitis should be treated with oral corticosteroids such as prednisolone weaning over 6–8 weeks. Patients with acute severe UC should be treated with high-dose IV corticosteroids such as methylprednisolone or hydrocortisone.
**Thiopurines**			
Azathioprine	Oral	Inhibition of proliferation of T and B lymphocytes. Decreased antibody production. Myelosuppression	Ulcerative colitis patients on maintenance therapy with high-dose 5-ASA, who required two or more courses of corticosteroids in the past year, or who become corticosteroid-dependent or refractory, require treatment with thiopurine.
**Monoclonal antibodies**			
Monoclonal antibodies anti-TNF-α		Inhibition of T lymphocytes differentiation. Induction of Treg lymphocytes. Decrease of macrophages activation. Diminution of NF-κβ pathways. Barrier improvement	Ulcerative colitis patients on maintenance therapy with high-dose 5-ASA, who required two or more courses of corticosteroids in the past year, or who become corticosteroid-dependent or refractory, require treatment with anti-TNF-α.
Infliximab	IV		Patients with acute severe UC failing to respond by day 3, as judged by a suitable scoring system, should be treated with rescue therapy in the form of intravenous infliximab for patients who have not previously failed thiopurine therapy
- Adalimumab - Golimumab	SC SC		
anti-integrin monoclonal antibody			
- Vedolizumab Human anti-α4β7 monoclonal antibody (anti-integrin)	IV	Inhibition of the adhesion of the T lymphocytes expressing the α4β7 to the molecule-1 of cellular adhesion of mucosal addressin (MAdCAM-1) mainly expressed on the intestinal endothelial cells. Decrease intestinal recruitment of T lymphocytes.	Ulcerative colitis patients on maintenance therapy with high-dose 5-ASA, who required two or more courses of corticosteroids in the past year, or who become corticosteroid-dependent or refractory, require treatment with Vedolizumab.
**Other** **treatments**			
Cyclosporin	IV	Inhibitory effects on T_H_ lymphocyte production of IL-2, and IFN-γ. Diminution of cytokines production (IL-3, 4 and 5, TNF-α). Diminution of T lymphocytes and B lymphocytes activation.	Patients with acute severe UC failing to respond by day 3, as judged by a suitable scoring system, should be treated with rescue therapy in the form of ciclosporin for patients who have not previously failed thiopurine therapy
Tofacitinib JAK inhibitor	Oral	Diminution of immune and inflammatory responses of JAK/STAT pathway.	Induction and maintenance of remission of ulcerative colitis in patients where anti-TNF treatment has failed. Ulcerative colitis patients on maintenance therapy with high-dose 5-ASA, who required two or more courses of corticosteroids in the past year, or who become corticosteroid-dependent or refractory, require treatment with tofacitinib.

5-ASA: 5-aminosalicylic acid; IL: Interleukin; IV: intravenous; JAK: Janus kinase; MAdCAM-1: Mucosal addressin cell adhesion molecule 1; NF-κB: nuclear factor-kappa B; SC: subcutaneous; STAT: signal transducer and activator of transcription; T_H_: T helper cell; TNF: Tumor necrosis factor; UC: ulcerative colitis.

**Table 3 microorganisms-08-01078-t003:** Evidences for the pathogenic role of CMV in ulcerative colitis (UC) patients.

Methods Used	Number of Patients Analyzed	Criteria Observed in Case of CMV Colitis when Compared to CMV Negative	Criteria and Publication
			**Severe disease**
pp65 antigenemia, IHC	47 UC	Higher endoscopic score	[122]
IHC	122 UC	Hospitalization	[123]
pp65 antigenemia	73 UC	Ulcers	[124]
CMV-pp65 antigenemia assay	118 UC	Delay to clinical remission	[125]
Heterogeneous (serology, histology, IHC, PCR)	72 UC	Higher disease flare-ups rate	[126]
Antigenemia, histology	222 UC	Hospitalization	[127]
Histology, pp65 antigenemia, IHC, PCR (in tissue or blood)	166 UC and 131 CD	Longer hospital stays	[108]
Histology, ISH, IHC	45 UC and 21 CD	Endoscopic ulcers	[128]
qPCR	40 IBD	Refractory disease	[80]
Histology, IHC	149 UC	Higher Mayo score and need for rescue therapy	[90]
IHC, qPCR	50 IBD	Refractory disease	[129]
Histology, IHC	56 UC	Longer hospital stays	[50]
Histology, IHC	257 UC	Deep ulcerations and higher disease activity, global poor outcome	[70]
qPCR in tissue	46 UC	Higher endoscopic score	[130]
qPCR in tissue	86 UC	Higher endoscopic score	[131]
PCR, IHC	239 UC	More severe disease	[63]
			**Increased mortality**
PCR, Serology (IgM), Histology	61 UC and 2 CD	Surgery, fatal outcome	[132]
IHC	95 UC	Lower hemoglobin and albumin levels, more intense histological inflammation	[85]
Not described	406,118 UC patients	Mortality, longer hospital stays, hospital charges	[1]
			**Surgery**
PCR, Serology (IgM), Histology	61 UC and 2 CD	Colectomy	[132]
Histology, IHC	77 UC	CMV found in surgical specimens of colectomy	[133]
Histology, IHC	126 UC	Colectomy	[134]
IHC	34 UC and 16 CD	Colectomy	[135]
Heterogeneous (serology, histology, IHC, PCR)	23 UC	Colectomy	[113]
Histology, IHC	26 UC and 17 CD	Colectomy	[72]
IHC	13 CD and 18 UC	Colectomy	[83]
qPCR in tissue	17 UC	Colectomy	[136]
IHC	13 UC	Colectomy	[137]
Heterogeneous (serology, histology, IHC, PCR)	72 UC	Colectomy	[126]
IHC, pp65 antigenemia	229 UC	Colectomy	[138]
IHC	77 UC	Surgical complications	[139]
Histology, IHC, qPCR in tissue	29 UC	Colectomy	[140]
IHC	95 UC	Colectomy	[85]
qPCR in tissue	108 UC	Protocolectomy	[141]
Histology, IHC	149 UC	Colectomy	[90]
Histology, IHC	56 UC	Colectomy	[50]
IHC	46 UC	Colectomy	[86]
Not described	406,118 UC patients	Colectomy	[1]
			**Resistance to immunosuppressive therapy**
pp65 antigenemia, IHC	47 UC	Steroid resistance	[122]
IHC	80 UC	Steroid resistance	[82]
Histology, IHC	77 UC	Steroid resistance	[133]
Histology, IHC	126 UC	Refractory disease	[134]
IHC	34 UC and 16 CD	Steroid resistance	[135]
Histology, IHC	49 UC and 23 CD	Steroid resistance	[142]
qPCR in tissue	42 UC	Steroid resistance Resistance to successive therapeutic lines	[44]
Histology, IHC, PCR	72 UC	Steroid resistance	[143]
pp65 antigenemia	187 UC	Steroid resistance	[144]
pp65 antigenemia	43 UC	Steroid resistance	[145]
qPCR in tissue	24 UC and 16 CD	Refractory disease	[80]
Histology, ISH, IHC	45 UC and 21 CD	Refractory disease	[128]
qPCR	35 UC	Steroid resistance and resistance to immunosuppressive treatment	[146]
qPCR	149 UC	Steroid resistance Resistance to successive therapeutic lines	[81]
Histology and IHC	56 UC	Steroid resistance	[50]
qPCR in tissue	46 UC	Higher corticosteroid doses	[130]
Histology and IHC	99 CD and 169 UC	Steroid dependence	[147]
Tissue PCR	52 UC patients	Steroid resistance	[43]
IHC or qPCR	80 UC patients	Steroid resistance	[96]

CD: Crohn disease; IHC: Immunohistochemistry; ISH: In situ hybridization; qPCR: Quantitative polymerase chain reaction; UC: ulcerative colitis.

**Table 4 microorganisms-08-01078-t004:** Predictive factors for CMV colitis in UC patients.

Country	Method for CMV detection	Population Studied	Predictive factors identified	OR in Multivariate Analysis	Publication
Japan	qPCR	86 patients with UC exacerbation	Age	1.08 [1.03–1.14]	[131]
			Endoscopic score	2.8 [0.65–12]	
			Combined immunosuppressants	7.44 [1.00–55.30]	
Germany	PCR in blood or PCR in tissue or IHC in tissue	239 UC patients	Disease activity (partial Mayo score)	1.37 [1.09–1.72]	[63]
			Use of steroids	2.43 [1.44–4.03]	
Israel	qPCR in tissue	28 UC patients	Use of steroids	14.5 [1.07–198]	[155]
			Fever > 38 °C	20 [1.2–330]	
Korea	H&E and IHC	149 patients	Use of steroids	3.30 [1.33–8.19]	[90]
			High Mayo score	1.58 [1.05–2.38]	
Japan	Agpp65 or H&E or IHC	149 UC	Use of systemic steroid dose at dose > 400 mg for 4 weeks before admission	26.70 [5.85–121.87]	[91]
			Punched-out ulcer	12.67 [4.21–38.14]	
Germany	IHC or tissue PCR	297 IBD patients	Age > 30 years	14.29 [2.89–118.57]	[108]
			Disease duration > 60 months	7.69 [1.80–45.41]	
			Blood leucocytes < 11/nl	4.49 [1.15–21.79]	
			Immunosuppressive therapy at admission	6.73 [1.67–35.63]	
United States	H&E and IHC or in situ hybridation	68 IBD patients	Age > 30 years	2.26 [1.02–5.03]	[128]
			Use of immunomodulators	1.95 [1.05–3.62]	
			Refractory disease	4.24 [2.21–8.11]	
Germany	qPCR	47 IBD patients	Use of steroid	7.1 [1.7–29.9]	[42]
			Use of calcineurin inhibitors	21.3 [2.4–188.7]	
			Use of 2 concurrent lines of immunosuppressive therapy	13.4 [3.2–56.1]	

CD: Crohn disease; IBD: Inflammatory Bowel Diseases; IHC: Immunohistochemistry; OR: Odd ratio with its 95% interval confidence; PCR: polymerase chain reaction (qualitative); qPCR: Quantitative polymerase chain reaction; UC: ulcerative colitis.

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
