# Peer review of "Cytomegalovirus and Inflammatory Bowel Diseases (IBD) with a Special Focus on the Link with Ulcerative Colitis (UC)"

_microorganisms, 2020, doi:10.3390/microorganisms8071078_

Round 1
Reviewer 1 Report
I have read with the big interest presented manuscript. In my opinion this comprehensive review can be very useful in the clinical practice and deserves the publication.
Author Response
We thank the reviewer for his/her appreciation of our manuscript.
Reviewer 2 Report
The review “Cytomegalovirus and inflammatory bowel diseases (IBD) with a special focus on the link with ulcerative colitis (UC)” by Jentzer et al. gives a detailed overview of biological characteristics of CMV and emphasizes the virological tools that are recommended for exploring CMV colitis during IBD. Moreover, the authors underline the interest of using ganciclovir for treating flare-ups associated to CMV colitis in UC patients.
In my opinion, the topic is relevant and of great interest. The Figures and Tables are adequate and are of help to follow the manuscript. Only minor modifications are suggested below.
It would be useful to cite and discuss these two works in the review: Ciccocioppo et al 2016 doi: 10.1007/s12026-015-8737-y and Nahar et al 2018 doi: 10.5217/ir.2018.16.1.90.
Line 232: some commas are missing in the sentence.
Line 434: change the title to English.
Author Response
We thank the reviewer for his/her appreciation of our manuscript.
It would be useful to cite and discuss these two works in the review: Ciccocioppo et al 2016 doi: 10.1007/s12026-015-8737-y and Nahar et al 2018 doi: 10.5217/ir.2018.16.1.90.
We added the lines 298 to 301 concerning the genotyping and included new references (including that of Nahar et al. 2018). We also added some data concerning the resistance to antiviral drugs (lines 302 to 304).
We added also some data concerning the detection of CMV DNA in stools (lines 312-314).
Concerning the works of the Italian team, the references numbered 80 and 131 were added in Table 3. Some words were added in the conclusion (line 467).
Line 232: some commas are missing in the sentence.
We added the 2 missing commas.
Line 434: change the title to English.
The line 434 is correctly written in English but we changed the line 474 (that was in French, our apologies for this mistake)